# Evidence Aggregation for Answer Re-Ranking in Open-Domain Question Answering

**Shuohang Wang**[1][*]**Mo Yu**[2][*]**Jing Jiang**[1]**,Wei Zhang**[2]**, Xiaoxiao Guo**[2]**, Shiyu Chang**[2]**, Zhiguo Wang**[2]
**Tim Klinger**[2]**, Gerald Tesauro**[2] **and Murray Campbell**[2]
[1]School of Information System, Singapore Management University
[2]AI Foundations - Learning, IBM Research AI
`shwang.2014@smu.edu.sg, yum@us.ibm.com, jingjiang@smu.edu.sg`

## ABSTRACT

A popular recent approach to answering open-domain questions is to first search for question-related passages and then apply reading comprehension models to extract answers. Existing methods usually extract answers from single passages independently. But some questions require a combination of evidence from across different sources to answer correctly. In this paper, we propose two models which make use of multiple passages to generate their answers. Both use an answer-reranking approach which reorders the answer candidates generated by an existing state-of-the-art QA model. We propose two methods, namely, *strength-based* re-ranking and *coverage-based* re-ranking, to make use of the aggregated evidence from different passages to better determine the answer. Our models have achieved state-of-the-art results on three public open-domain QA datasets: Quasar-T, SearchQA and the open-domain version of TriviaQA, with about 8 percentage points of improvement over the former two datasets.

## 1 INTRODUCTION

Open-domain question answering (QA) aims to answer questions from a broad range of domains by effectively marshalling evidence from large open-domain knowledge sources. Such resources can be Wikipedia (Chen et al., 2017), the whole web (Ferrucci et al., 2010), structured knowledge bases (Berant et al., 2013; Yu et al., 2017) or combinations of the above (Baudiš & Šedivỳ, 2015). Recent work on open-domain QA has focused on using unstructured text retrieved from the web to build machine comprehension models (Chen et al., 2017; Dhingra et al., 2017b; Wang et al., 2017). These studies adopt a two-step process: an information retrieval (IR) model to coarsely select passages relevant to a question, followed by a reading comprehension (RC) model (Wang & Jiang, 2017; Seo et al., 2017; Chen et al., 2017) to infer an answer from the passages. These studies have made progress in bringing together evidence from large data sources, but they predict an answer to the question with only a single retrieved passage at a time. However, answer accuracy can often be improved by using multiple passages. In some cases, the answer can only be determined by combining multiple passages.

In this paper, we propose a method to improve open-domain QA by explicitly aggregating evidence from across multiple passages. Our method is inspired by two notable observations from previous open-domain QA results analysis:

- First, compared with incorrect answers, the correct answer is often suggested by more passages repeatedly. For example, in Figure 1(a), the correct answer "*danny boy*" has more passages providing evidence relevant to the question compared to the incorrect one. This observation can be seen as multiple passages collaboratively enhancing the evidence for the correct answer.

- Second, sometimes the question covers multiple answer aspects, which spreads over multiple passages. In order to infer the correct answer, one has to find ways to aggregate those multiple passages in an effective yet sensible way to try to cover all aspects. In Figure 1(b), for example,

---

[*]Equal contribution.

**Question1:** What is the more popular name for the londonderry air?

**A1: tune from county**

**P1**: the best known title for this melody is londonderry air -lrb- sometimes also called the **tune from county** derry -rrb- .

**A2: danny boy**

**P1**: londonderry air words : this melody is more commonly known with the words `` **danny boy** ''

**P2**: londonderry air **danny boy** music ftse london i love you .

**P3**: **danny boy** limavady is most famous for the tune londonderry air collected by jane ross in the mid-19th century from a local fiddle player .

**P4**: it was here that jane ross noted down the famous londonderry air -lrb- ` **danny boy** ' -rrb- from a passing fiddler .

(a)

**Question2:** Which physicist , mathematician and astronomer discovered the first 4 moons of Jupiter

**A1: Isaac Newton**

**P1**: **Sir Isaac Newton** was an English physicist , mathematician , astronomer , natural philosopher , alchemist and theologian …

**P2**: **Sir Isaac Newton** was an English mathematician, astronomer, and physicist who is widely recognized as one of the most influential scientists …

**Question2:** Which physicist , mathematician and astronomer discovered the first 4 moons of Jupiter

**A2: Galileo Galilei**

**P1**: **Galileo Galilei** was an Italian physicist , mathematician , astronomer , and philosopher who played a major role in the Scientific Revolution .

**P2**: **Galileo Galilei** is credited with discovering the first four moons of Jupiter .

(b)

Figure 1: Two examples of questions and candidate answers. (a) A question benefiting from the repetition of evidence. Correct answer A2 has multiple passages that could support A2 as answer. The wrong answer A1 has only a single supporting passage. (b) A question benefiting from the union of multiple pieces of evidence to support the answer. The correct answer A2 has evidence passages that can match both the first half and the second half of the question. The wrong answer A1 has evidence passages covering only the first half.

the correct answer "*Galileo Galilei*" at the bottom has passages P1, "*Galileo was a physicist ...*" and P2, "*Galileo discovered the first 4 moons of Jupiter*", mentioning two pieces of evidence to match the question. In this case, the aggregation of these two pieces of evidence can help entail the ground-truth answer "*Galileo Galilei*". In comparison, the incorrect answer "*Isaac Newton*" has passages providing partial evidence on only "*physicist, mathematician and astronomer*". This observation illustrates the way in which multiple passages may provide ***complementary*** evidence to better infer the correct answer to a question.

To provide more accurate answers for open-domain QA, we hope to make better use of multiple passages for the same question by aggregating both the strengthened and the complementary evidence from all the passages. We formulate the above evidence aggregation as an answer re-ranking problem. Re-ranking has been commonly used in NLP problems, such as in parsing and translation, in order to make use of high-order or global features that are too expensive for decoding algorithms (Collins & Koo, 2005; Shen et al., 2004; Huang, 2008; Dyer et al., 2016). Here we apply the idea of re-ranking; for each answer candidate, we efficiently incorporate global information from multiple pieces of textual evidence without significantly increasing the complexity of the prediction of the RC model. Specifically, we first collect the top-$K$ candidate answers based on their probabilities computed by a standard RC/QA system, and then we use two proposed re-rankers to re-score the answer candidates by aggregating each candidate's evidence in different ways. The re-rankers are:

- A ***strength-based re-ranker***, which ranks the answer candidates according to how often their evidence occurs in different passages. The re-ranker is based on the first observation if an answer candidate has multiple pieces of evidence, and each passage containing some evidence tends to predict the answer with a relatively high score (although it may not be the top score), then the candidate is more likely to be correct. The passage count of each candidate, and the aggregated probabilities for the candidate, reflect how strong its evidence is, and thus in turn suggest how likely the candidate is the corrected answer.

- A ***coverage-based re-ranker***, which aims to rank an answer candidate higher if the union of all its contexts in different passages could cover more aspects included in the question. To achieve this, for each answer we concatenate all the passages that contain the answer together. The result is a new context that aggregates all the evidence necessary to entail the answer for the question. We then treat the new context as one sequence to represent the answer, and build an attention-based match-LSTM model (Wang & Jiang, 2017) between the sequence and the question to measure how well the new aggregated context could entail the question.

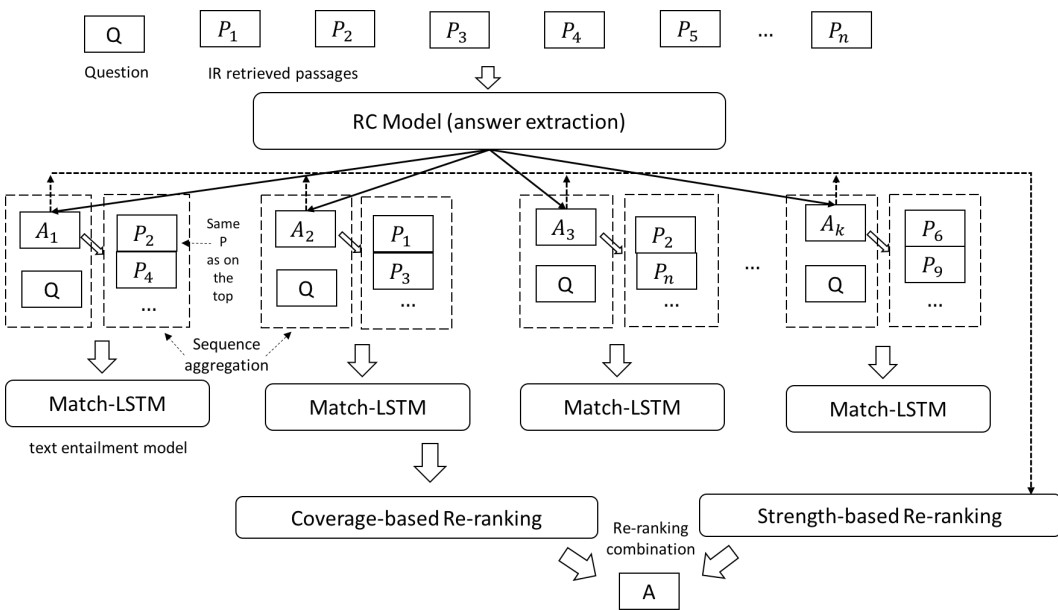

Figure 2: An overview of the full re-ranker. It consists of strength-based and coverage-based re-ranking.

Overall, our contributions are as follows: 1) We propose a re-ranking-based framework to make use of the evidence from multiple passages in open-domain QA, and two re-rankers, namely, a strength-based re-ranker and a coverage-based re-ranker, to perform evidence aggregation in existing open-domain QA datasets. We find the second re-ranker performs better than the first one on two of the three public datasets. 2) Our proposed approach leads to the state-of-the-art results on three different datasets (**Quasar-T** (Dhingra et al., 2017b), **SearchQA** (Dunn et al., 2017) and **TriviaQA** (Joshi et al., 2017)) and outperforms previous state of the art by large margins. In particular, we achieved up to 8% improvement on F1 on both Quasar-T and SearchQA compared to the previous best results.

## 2 METHOD

Given a question $\mathbf{q}$, we are trying to find the correct answer $\mathbf{a}^g$ to $\mathbf{q}$ using information retrieved from the web. Our method proceeds in two phases. First, we run an IR model (with the help of a search engine such as google or bing) to find the top-$N$ web passages $\mathbf{p}_1, \mathbf{p}_2, \ldots, \mathbf{p}_N$ most related to the question. Then a reading comprehension (RC) model is used to extract the answer from these passages. This setting is different from standard reading comprehension tasks (e.g. (Rajpurkar et al., 2016)), where a single fixed passage is given, from which the answer is to be extracted. When developing a reading comprehension system, we can use the specific positions of the answer sequence in the given passage for training. By contrast, in the open-domain setting, the RC models are usually trained under distant supervision (Chen et al., 2017; Dhingra et al., 2017b; Joshi et al., 2017). Specifically, since the training data does not have labels indicating the positions of the answer spans in the passages, during the training stage, the RC model will match all passages that contain the ground-truth answer with the question one by one. In this paper we apply an existing RC model called $R^3$ (Wang et al., 2017) to extract these candidate answers.

After the candidate answers are extracted, we aggregate evidence from multiple passages by re-ranking the answer candidates. Given a question $\mathbf{q}$, suppose we have a baseline open-domain QA system that can generate the top-$K$ answer candidates $\mathbf{a}_1, \ldots, \mathbf{a}_K$, each being a text span in some passage $\mathbf{p}_i$. The goal of the re-ranker is to rank this list of candidates so that the top-ranked candidates are more likely to be the correct answer $\mathbf{a}^g$. With access to these additional features, the re-ranking step has the potential to prioritize answers not easily discoverable by the base system alone. We investigate two re-ranking strategies based on *evidence strength* and *evidence coverage*. An overview of our method is shown in Figure 2.

## 2.1 EVIDENCE AGGREGATION FOR STRENGTH-BASED RE-RANKER

In open-domain QA, unlike the standard RC setting, we have more passages retrieved by the IR model and the ground-truth answer may appear in different passages, which means different answer spans may correspond to the same answer. To exploit this property, we provide two features to further re-rank the top-$K$ answers generated by the RC model.

**Measuring Strength by Count**  This method is based on the hypothesis that the more passages that entail a particular answer, the stronger the evidence for that answer and the higher it should be ranked. To implement this we count the number of occurrences of each answer in the top-$K$ answer spans generated by the baseline QA model and return the answer with the highest count.

**Measuring Strength by Probability**  Since we can get the probability of each answer span in a passages based on the RC model, we can also sum up the probabilities of the answer spans that are referring to the same answer. In this method, the answer with the highest probability is the final prediction [1]. In the re-ranking scenario, it is not necessary to exhaustively consider all the probabilities of all the spans in the passages, as there may be a large number of different answer spans and most of them are irrelevant to the ground-truth answer.

**Remark**: Note that neither of the above methods require any training. Both just take the candidate predictions from the baseline QA system and perform counting or probability calculations. At test time, the time complexity of strength-based re-ranking is negligible.

## 2.2 EVIDENCE AGGREGATION FOR COVERAGE-BASED RE-RANKER

Consider Figure 1 where the two answer candidates both have evidence matching the first half of the question. Note that only the correct answer has evidence that could also match the second half. In this case, the strength-based re-ranker will treat both answer candidates the same due to the equal amount of supporting evidence, while the second answer has complementary evidence satisfying all aspects of the question. To handle this case, we propose a *coverage-based re-ranker* that ranks the answer candidates according to how well the union of their evidence from different passages covers the question.

In order to take the union of evidence into consideration, we first concatenate the passages containing the answer into a single "pseudo passage" then measure how well this passage entails the answer for the question. As in the examples shown in Figure 1(b), we hope the textual entailment model will reflect (i) how each aspect of the question is matched by the union of multiple passages; and (ii) whether all the aspects of the question can be matched by the union of multiple passages. In our implementation an "aspect" of the question is a hidden state of a bi-directional LSTM (Hochreiter & Schmidhuber, 1997). The match-LSTM (Wang & Jiang, 2016) model is one way to achieve the above effect in entailment. Therefore we build our coverage-based re-ranker on top of the concatenated pseudo passages using the match-LSTM. The detailed method is described below.

**Passage Aggregation**  We consider the top-$K$ answers, $\mathbf{a}_1, \ldots, \mathbf{a}_K$, provided by the baseline QA system. For each answer $\mathbf{a}_k, k \in [1, K]$, we concatenate all the passages that contain $\mathbf{a}_k$, $\{\mathbf{p}_n | \mathbf{a}_k \in \mathbf{p}_n, n \in [1, N]\}$, to form ***the union passage*** $\hat{\mathbf{p}}_k$. Our further model is to identify which union passage, e.g., $\hat{\mathbf{p}}_k$, could better entail its answer, e.g., $\mathbf{a}_k$, for the question.

**Measuring Aspect(Word)-Level Matching**  As discussed earlier, the first mission of the coverage-based re-ranker is to measure how each aspect of the question is matched by the union of multiple passages. We achieve this with word-by-word attention followed by a comparison module.

First, we write the answer candidate $\mathbf{a}$, question $\mathbf{q}$ and the union passage $\hat{\mathbf{p}}$ of $\mathbf{a}$ as matrices $\mathbf{A}, \mathbf{Q}, \hat{\mathbf{P}}$, with each column being the embedding of a word in the sequence. We then feed them to the bi-directional LSTM as follows:

$$\mathbf{H}^a = \text{BiLSTM}(\mathbf{A}), \quad \mathbf{H}^q = \text{BiLSTM}(\mathbf{Q}), \quad \mathbf{H}^p = \text{BiLSTM}(\hat{\mathbf{P}}), \tag{1}$$

---

[1]This is an extension of the Attention Sum method in (Kadlec et al., 2016) from single-token answers to phrase answers.

where $\mathbf{H}^{\mathrm{a}} \in \mathbb{R}^{l \times A}$, $\mathbf{H}^{\mathrm{q}} \in \mathbb{R}^{l \times Q}$ and $\mathbf{H}^{\mathrm{p}} \in \mathbb{R}^{l \times P}$ are the hidden states for the answer candidate, question and passage respectively; $l$ is the dimension of the hidden states, and $A$, $Q$ and $P$ are the length of the three sequences, respectively.

Next, we enhance the question representation $\mathbf{H}^{\mathrm{q}}$ with $\mathbf{H}^{\mathrm{a}}$:

$$\mathbf{H}^{\mathrm{aq}} = [\mathbf{H}^{\mathrm{a}}; \mathbf{H}^{\mathrm{q}}], \qquad (2)$$

where $[\cdot;\cdot]$ is the concatenation of two matrices in row and $\mathbf{H}^{\mathrm{aq}} \in \mathbb{R}^{l \times (A+Q)}$. As most of the answer candidates do not appear in the question, this is for better matching with the passage and finding more answer-related information from the passage.[2] Now we can view each *aspect* of the question as a column vector (i.e. a hidden state at each word position in the answer-question concatenation) in the enhanced question representation $\mathbf{H}^{\mathrm{aq}}$. Then the task becomes to measure how well each column vector can be matched by the union passage; and we achieve this by computing the attention vector Parikh et al. (2016) for each hidden state of sequences $\mathbf{a}$ and $\mathbf{q}$ as follows:

$$\alpha = \mathrm{SoftMax}\left((\mathbf{H}^{\mathrm{p}})^{\mathsf{T}}\mathbf{H}^{\mathrm{aq}}\right), \quad \overline{\mathbf{H}}^{\mathrm{aq}} = \mathbf{H}^{\mathrm{p}}\alpha, \qquad (3)$$

where $\alpha \in \mathbb{R}^{P \times (A+Q)}$ is the attention weight matrix which is normalized in column through softmax. $\overline{\mathbf{H}}^{\mathrm{aq}} \in \mathbb{R}^{l \times (A+Q)}$ are the attention vectors for each word of the answer and the question by weighted summing all the hidden states of the passage $\hat{\mathbf{p}}$. Now in order to see whether the aspects in the question can be matched by the union passage, we use the following matching function:

$$\mathbf{M} = \mathrm{ReLU}\left(\mathbf{W}^{\mathrm{m}}\begin{bmatrix}\mathbf{H}^{\mathrm{aq}} \odot \overline{\mathbf{H}}^{\mathrm{aq}} \\ \mathbf{H}^{\mathrm{aq}} - \overline{\mathbf{H}}^{\mathrm{aq}} \\ \mathbf{H}^{\mathrm{aq}} \\ \overline{\mathbf{H}}^{\mathrm{aq}}\end{bmatrix} + \mathbf{b}^{\mathrm{m}} \otimes \mathbf{e}_{(A+Q)}\right), \qquad (4)$$

where $\cdot \otimes \mathbf{e}_{(A+Q)}$ is to repeat the vector (or scalar) on the left $A + Q$ times; $(\cdot \odot \cdot)$ and $(\cdot - \cdot)$ are the element-wise operations for checking whether the word in the answer and question can be matched by the evidence in the passage. We also concatenate these matching representations with the hidden state representations $\mathbf{H}^{\mathrm{aq}}$ and $\overline{\mathbf{H}}^{\mathrm{aq}}$, so that the lexical matching representations are also integrated into the the final aspect-level matching representations[3] $\mathbf{M} \in \mathbb{R}^{2l \times (A+Q)}$, which is computed through the non-linear transformation on four different representations with parameters $\mathbf{W}^{\mathrm{m}} \in \mathbb{R}^{2l \times 4l}$ and $b^{\mathrm{m}} \in \mathbb{R}^{l}$.

**Measuring the Entire Question Matching** Next, in order to measure how the entire question is matched by the union passage $\hat{\mathbf{p}}$ by taking into consideration of the matching result at each aspect, we add another bi-directional LSTM on top of it to aggregate the aspect-level matching information [4]:

$$\mathbf{H}^{\mathrm{m}} = \mathrm{BiLSTM}(\mathbf{M}), \quad \mathbf{h}^{\mathrm{s}} = \mathrm{MaxPooling}(\mathbf{H}^{\mathrm{m}}), \qquad (5)$$

where $\mathbf{H}^{\mathrm{m}} \in \mathbb{R}^{l \times (A+Q)}$ is to denote all the hidden states and $\mathbf{h}^{\mathrm{s}} \in \mathbb{R}^{l}$, the max of pooling of each dimension of $\mathbf{H}^{\mathrm{m}}$, is the entire matching representation which reflects how well the evidences in questions could be matched by the union passage.

---

[2]Besides concatenating $\mathbf{H}^{\mathrm{q}}$ with $\mathbf{H}^{\mathrm{a}}$, there are other ways to make the matching process be aware of an answer's positions in the passage, e.g. replacing the answer spans in the passage to a special token like in Yih et al. (2015). We tried this approach, which gives similar but no better results, so we keep the concatenation in this paper. We leave the study of the better usage of answer position information for future work.

[3]Concatenating $\mathbf{H}^{\mathrm{aq}}$ and $\overline{\mathbf{H}}^{\mathrm{aq}}$ could help the question-level matching (see Eq. 5 in the next paragraph) by allowing the BiLSTM learn to distinguish the effects of the element-wise comparison vectors with the original lexical information. If we only use the element-wise comparison vectors, the model may not be able to know what the matched words/contexts are.

[4]Note that we use LSTM here to capture the conjunction information (the dependency) among aspects, i.e. how all the aspects are jointly matched. In comparison simple pooling methods will treat the aspects independently. Low-rank tensor inspired neural architectures (e.g., Lei et al. (2017)) could be another choice and we will investigate them in future work.

**Re-ranking Objective Function**   Our re-ranking is based on the entire matching representation. For each candidate answer $\mathbf{a}_k, k \in [1, K]$, we can get a matching representation $\mathbf{h}_k^s$ between the answer $\mathbf{a}_k$, question $\mathbf{q}$ and the union passage $\hat{\mathbf{p}}_k$ through Eqn. (1-5). Then we transform all representations into scalar values followed by a normalization process for ranking:

$$\mathbf{R} = \mathrm{Tanh}\left(\mathbf{W}^r[\mathbf{h}_1^s; \mathbf{h}_2^s; ...; \mathbf{h}_K^s] + \mathbf{b}^r \otimes \mathbf{e}_K\right), \quad \mathbf{o} = \mathrm{Softmax}(\mathbf{w}^o\mathbf{R} + b^o \otimes \mathbf{e}_K), \qquad (6)$$

where we concatenate the match representations for each answer in row through $[\cdot; \cdot]$, and do a non-linear transformation by parameters $\mathbf{W}^r \in \mathbb{R}^{l \times l}$ and $\mathbf{b}^r \in \mathbb{R}^l$ to get hidden representation $\mathbf{R} \in \mathbb{R}^{l \times K}$. Finally, we map the transformed matching representations into scalar values through parameters $\mathbf{w}^o \in \mathbb{R}^l$ and $\mathbf{w}^o \in \mathbb{R}$. $\mathbf{o} \in \mathbb{R}^K$ is the normalized probability for the candidate answers to be ground-truth. Due to the aliases of the ground-truth answer, there may be multiple answers in the candidates are ground-truth, we use KL distance as our objective function:

$$\sum_{k=1}^K y_k \left(\log(y_k) - \log(o_k)\right), \qquad (7)$$

where $y_k$ indicates whether $\mathbf{a}_k$ the ground-truth answer or not and is normalized by $\sum_{k=1}^K y_k$ and $o_k$ is the ranking output of our model for $\mathbf{a}_k$.

## 2.3   COMBINATION OF DIFFERENT TYPES OF AGGREGATIONS

Although the coverage-based re-ranker tries to deal with more difficult cases compared to the strength-based re-ranker, the strength-based re-ranker works on more common cases according to the distributions of most open-domain QA datasets. We can try to get the best of both worlds by combining the two approaches. The ***full re-ranker*** is a weighted combination of the outputs of the above different re-rankers without further training. Specifically, we first use softmax to re-normalize the top-5 answer scores provided by the two strength-based rankers and the one coverage-based re-ranker; we then weighted sum up the scores for the same answer and select the answer with the largest score as the final prediction.

## 3   EXPERIMENTAL SETTINGS

We conduct experiments on three publicly available open-domain QA datasets, namely, **Quasar-T** (Dhingra et al., 2017b), **SearchQA** (Dunn et al., 2017) and **TriviaQA** (Joshi et al., 2017). These datasets contain passages retrieved for all questions using a search engine such as Google or Bing. We do not retrieve more passages but use the provided passages only.

### 3.1   DATASETS

The statistics of the three datasets are shown in Table 1.

**Quasar-T** [5] (Dhingra et al., 2017b) is based on a trivia question set. The data set makes use of the "Lucene index" on the ClueWeb09 corpus. For each question, 100 unique sentence-level passages were collected. The human performance is evaluated in an open-book setting, i.e., the human subjects had access to the same passages retrieved by the IR model and tried to find the answers from the passages.

**SearchQA** [6] (Dunn et al., 2017) is based on Jeopardy! questions and uses Google to collect about 50 web page snippets as passages for each question. The human performance is evaluated in a similar way to the Quasar-T dataset.

**TriviaQA (Open-Domain Setting)** [7] (Joshi et al., 2017) collected trivia questions coming from 14 trivia and quiz-league websites, and makes use of the Bing Web search API to collect the top 50

---

[5] https://github.com/bdhingra/quasar
[6] https://github.com/nyu-dl/SearchQA
[7] http://nlp.cs.washington.edu/triviaqa/data/triviaqa-unfiltered.tar.gz

|  | #q(train) | #q(dev) | #q(test) | #p | #p(truth) | #p(aggregated) |
|---|---|---|---|---|---|---|
| Quasar-T | 28,496 | 3,000 | 3,000 | 100 | 14.8 | 5.2 |
| SearchQA | 99,811 | 13,893 | 27,247 | 50 | 16.5 | 5.4 |
| TriviaQA | 66,828 | 11,313 | 10,832 | 100 | 16.0 | 5.6 |

Table 1: Statistics of the datasets. #q represents the number of questions for training (not counting the questions that don't have ground-truth answer in the corresponding passages for training set), development, and testing datasets. #p is the number of passages for each question. For TriviaQA, we split the raw documents into sentence level passages and select the top 100 passages based on the its overlaps with the corresponding question. #p(golden) means the number of passages that contain the ground-truth answer in average. #p(aggregated) is the number of passages we aggregated in average for top 10 candidate answers provided by RC model.

webpages most related to the questions. We focus on the open domain setting (the unfiltered passage set) of the dataset [8] and our model uses all the information retrieved by the IR model.

## 3.2 BASELINES

Our baseline models [9] include the following: GA (Dhingra et al., 2017a;b), a reading comprehension model with gated-attention; BiDAF (Seo et al., 2017), a RC model with bidirectional attention flow; AQA (Buck et al., 2017), a reinforced system learning to aggregate the answers generated by the re-written questions; $R^3$ (Wang et al., 2017), a reinforced model making use of a ranker for selecting passages to train the RC model. As $R^3$ is the first step of our system for generating candidate answers, the improvement of our re-ranking methods can be directly compared to this baseline.

TriviaQA does not provide the leaderboard under the open-domain setting. As a result, there is no public baselines in this setting and we only compare with the $R^3$ baseline.[10]

## 3.3 IMPLEMENTATION DETAILS

We first use a pre-trained $R^3$ model (Wang et al., 2017), which gets the state-of-the-art performance on the three public datasets we consider, to generate the top 50 candidate spans for the training, development and test datasets, and we use them for further ranking. During training, if the ground-truth answer does not appear in the answer candidates, we will manually add it into the answer candidate list.

For the coverage-based re-ranker, we use Adam (Kingma & Ba, 2015) to optimize the model. Word embeddings are initialized by GloVe (Pennington et al., 2014) and are not updated during training. We set all the words beyond Glove as zero vectors. We set $l$ to 300, batch size to 30, learning rate to 0.002. We tune the dropout probability from 0 to 0.5 and the number of candidate answers for re-ranking ($K$) in [3, 5, 10] [11].

## 4 RESULTS AND ANALYSIS

In this section, we present results and analysis of our different re-ranking methods on the three different public datasets.

---

[8]Despite the open-domain QA data provided, the leaderboard of TriviaQA focuses on evaluation of RC models over filtered passages that is guaranteed to contain the correct answers (i.e. more like closed-domain setting). The evaluation is also passage-wise, different from the open-domain QA setting.

[9] Most of the results of different models come from the public paper. While we re-run model $R^3$ (Wang et al., 2017) based on the authors' source code and extend the model to the datasets of SearchQA and TriviaQA datasets.

[10]To demonstrate that $R^3$ serves as a strong baseline on the TriviaQA data, we generate the $R^3$ results following the leaderboard setting. The results showed that $R^3$ achieved F1 56.0, EM 50.9 on Wiki domain and F1 68.5, EM 63.0 on Web domain, which is competitive to the state-of-the-arts. This confirms that $R^3$ is a competitive baseline when extending the TriviaQA questions to open-domain setting.

[11]Our code will be released under `https://github.com/shuohangwang/mprc`.

| | Quasar-T | | SearchQA | | TriviaQA (open) | |
|---|---|---|---|---|---|---|
| | EM | F1 | EM | F1 | EM | F1 |
| GA (Dhingra et al., 2017a) | 26.4 | 26.4 | - | - | - | - |
| BiDAF (Seo et al., 2017) | 25.9 | 28.5 | 28.6 | 34.6 | - | - |
| AQA (Buck et al., 2017) | - | - | 40.5 | 47.4 | - | - |
| $R^3$ (Wang et al., 2017) | 35.3 | 41.7 | 49.0 | 55.3 | 47.3 | 53.7 |
| Baseline Re-Ranker (BM25) | 33.6 | 45.2 | 51.9 | 60.7 | 44.6 | 55.7 |
| Our Full Re-Ranker | **42.3** | **49.6** | **57.0** | **63.2** | **50.6** | **57.3** |
|     Strength-Based Re-Ranker (Probability) | 36.1 | 42.4 | 50.4 | 56.5 | 49.2 | 55.1 |
|     Strength-Based Re-Ranker (Counting) | 37.1 | 46.7 | 54.2 | 61.6 | 46.1 | 55.8 |
|     Coverage-Based Re-Ranker | 40.6 | 49.1 | 54.1 | 61.4 | 50.0 | 57.0 |
| Human Performance | 51.5 | 60.6 | 43.9 | - | - | - |

Table 2: Experiment results on three open-domain QA test datasets: Quasar-T, SearchQA and Trivi-aQA (open-domain setting). EM: Exact Match. Full Re-ranker is the combination of three different re-rankers.

## 4.1 Overall Results

The performance of our models is shown in Table 2. We use F1 score and Exact Match (EM) as our evaluation metrics [12]. From the results, we can clearly see that the full re-ranker, the combination of different re-rankers, significantly outperforms the previous best performance by a large margin, especially on Quasar-T and SearchQA. Moreover, our model is much better than the human performance on the SearchQA dataset. In addition, we see that our coverage-based re-ranker achieves consistently good performance on the three datasets, even though its performance is marginally lower than the strength-based re-ranker on the SearchQA dataset.

## 4.2 Analysis

In this subsection, we analyze the benefits of our re-ranking models.

**BM25 as an alternative coverage-based re-ranker** We use the classical BM25 retrieval model (Robertson et al., 2009) to re-rank the aggregated passages the same way as the coverage-based re-ranker, where the IDF values are first computed from the raw passages before aggregation. From the results in Table 2, we see that the BM25-based re-ranker improves the F1 scores compared with the $R^3$ model, but it is still lower than our coverage-based re-ranker with neural network models. Moreover, with respect to EM scores, the BM25-based re-ranker sometimes gives lower performance. We hypothesize that there are two reasons behind the relatively poor performance of BM25. First, because BM25 relies on a bag-of-words representation, context information is not taken into consideration and it cannot model the phrase similarities. Second, shorter answers tend to be preferred by BM25. For example, in our method of constructing pseudo-passages, when an answer sequence $A$ is a subsequence of another answer sequence $B$, the pseudo passage of $A$ is always a superset of the pseudo passage of $B$ that could better cover the question. Therefore the F1 score could be improved but the EM score sometimes becomes worse.

**Re-ranking performance versus answer lengths and question types** Figure 3 decomposes the performance according to the length of the ground truth answers and the types of questions on TriviaQA and Quasar-T. We do not include the analysis on SearchQA because, for the Jeopardy! style questions, it is more difficult to distinguish the questions types, and the range of answer lengths is narrower.

Our results show that the coverage-based re-ranker outperforms the baseline in different lengths of answers and different types of questions. The strength-based re-ranker (counting) also gives improvement but is less stable across different datasets, while the strength-based re-ranker (probability)

---

[12]Our evaluation is based on the tool from SQuAD (Rajpurkar et al., 2016).

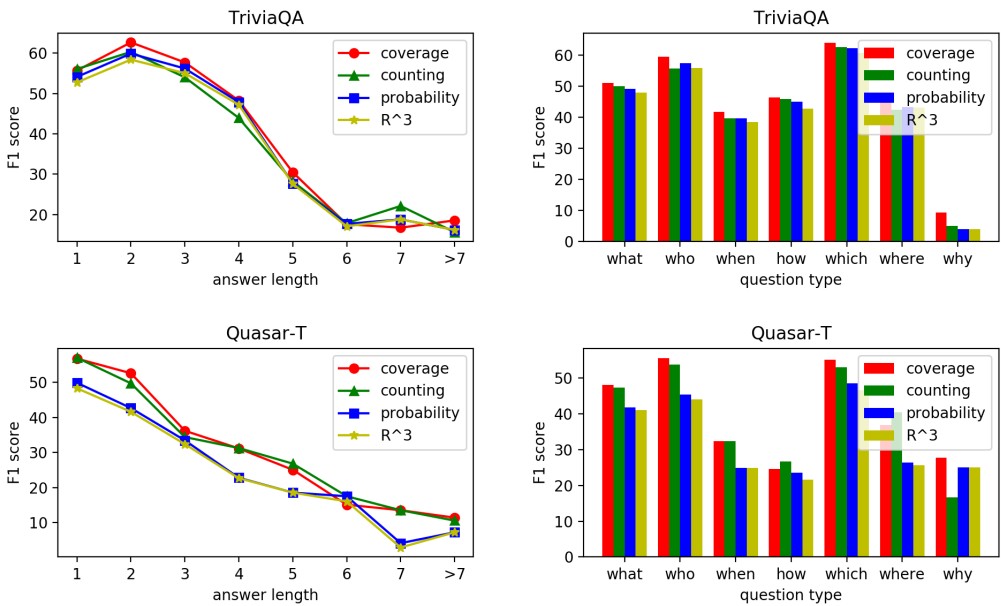

Figure 3: Performance decomposition according to the length of answers and the question types.

| | Quasar-T | | SearchQA | | TriviaQA (open) | |
|---|---|---|---|---|---|---|
| **Top-K** | **EM** | **F1** | **EM** | **F1** | **EM** | **F1** |
| 1 | 35.1 | 41.6 | 51.2 | 57.3 | 47.6 | 53.5 |
| 3 | 46.2 | 53.5 | 63.9 | 68.9 | 54.1 | 60.4 |
| 5 | 51.0 | 58.9 | 69.1 | 73.9 | 58.0 | 64.5 |
| 10 | 56.1 | 64.8 | 75.5 | 79.6 | 62.1 | 69.0 |

Table 3: The upper bound (recall) of the Top-K answer candidates generated by the baseline $R^3$ system (on dev set), which indicates the potential of the coverage-based re-ranker.

tends to have results and trends that are close to the baseline curves, which is probably because the method is dominated by the probabilities predicted by the baseline.

The coverage-based re-ranker and the strength-based re-ranker (counting) have similar trends on most of the question types. The only exception is that the strength-based re-ranker performs significantly worse compared to the coverage-based re-ranker on the "*why*" questions. This is possibly because those questions usually have non-factoid answers, which are less likely to have exactly the same text spans predicted on different passages by the baseline.

**Potential improvement of re-rankers** Table 3 shows the percentage of times the correct answer is included in the top-$K$ answer predictions of the baseline $R^3$ method. More concretely, the scores are computed by selecting the answer from the top-$K$ predictions with the best EM/F1 score. Therefore the final top-$K$ EM and F1 can be viewed as the recall or an upper bound of the top-$K$ predictions. From the results, we can see that although the top-1 prediction of $R^3$ is not very accurate, there is high probability that a top-$K$ list with small $K$ could cover the correct answer. This explains why our re-ranking approach achieves large improvement. Also by comparing the upper bound performance of top-5 and our re-ranking performance in Table 2, we can see there is still a clear gap of about 10% on both datasets and on both F1 and EM, showing the great potential improvement for the re-ranking model in future work.

**Effect of the selection of $K$ for the coverage-based re-ranker** As shown in Table 3, as $K$ ranges from 1 to 10, the recall of top-$K$ predictions from the baseline $R^3$ system increases significantly.

| Candidate Set | Re-Ranker Results | | Upper Bound | |
|---|---|---|---|---|
| | EM | F1 | EM | F1 |
| top-3 | 40.5 | 47.8 | 46.2 | 53.5 |
| top-5 | **41.8** | 50.1 | 51.0 | 58.9 |
| top-10 | 41.3 | **50.8** | 56.1 | 64.8 |

Table 4: Results of running coverage-based re-ranker on different number of the top-$K$ answer candidates on Quasar-T (dev set).

| Candidate Set | Re-Ranker Results | | Upper Bound | |
|---|---|---|---|---|
| | EM | F1 | EM | F1 |
| top-10 | **37.9** | 46.1 | 56.1 | 64.8 |
| top-50 | 37.8 | **47.8** | 64.1 | 74.1 |
| top-100 | 36.4 | 47.3 | 66.5 | 77.1 |
| top-200 | 33.7 | 45.8 | 68.7 | 79.5 |

Table 5: Results of running strength-based re-ranker (counting) on different number of top-$K$ answer candidates on Quasar-T (dev set).

Ideally, if we use a larger $K$, then the candidate lists will be more likely to contain good answers. At the same time, the lists to be ranked are longer thus the re-ranking problem is harder. Therefore, there is a trade-off between the coverage of rank lists and the difficulty of re-ranking; and selecting an appropriate $K$ becomes important. Table 4 shows the effects of $K$ on the performance of coverage-based re-ranker. We train and test the coverage-based re-ranker on the top-$K$ predictions from the baseline, where $K \in \{3, 5, 10\}$. The upper bound results are the same ones from Table 3. The results show that when $K$ is small, like $K$=3, the performance is not very good due to the low coverage (thus low upper bound) of the candidate list. With the increase of $K$, the performance becomes better, but the top-5 and top-10 results are on par with each other. This is because the higher upper bound of top-10 results counteracts the harder problem of re-ranking longer lists. Since there is no significant advantage of the usage of $K$=10 while the computation cost is higher, we report all testing results with $K$=5.

**Effect of the selection of $K$ for the strength-based re-ranker**    Similar to Table 4, we conduct experiments to show the effects of $K$ on the performance of the strength-based re-ranker. We run the strength-based re-ranker (counting) on the top-$K$ predictions from the baseline, where $K \in \{10, 50, 100, 200\}$. We also evaluate the upper bound results for these $K$s. Note that the strength-based re-ranker is very fast and the different values of $K$ do not affect the computation speed significantly compared to the other QA components.

The results are shown in Table 5, where we achieve the best results when $K$=50. The performance drops significantly when $K$ increases to 200. This is because the ratio of incorrect answers increases notably, making incorrect answers also likely to have high counts. When $K$ is smaller, such incorrect answers appear less because statistically they have lower prediction scores. We report all testing results with $K$=50.

**Examples**    Table 6 shows an example from Quasar-T where the re-ranker successfully corrected the wrong answer predicted by the baseline. This is a case where the coverage-based re-ranker helped: the correct answer "*Sesame Street*" has evidence from different passages that covers the aspects "*Emmy Award*" and "*children 's television shows*". Although it still does not fully match all the facts in the question, it still helps to rank the correct answer higher than the top-1 prediction "*Great Dane*" from the R[3] baseline, which only has evidence covering "*TV*" and "*1969*" in the question.

| Q: | Which children 's TV programme , which first appeared in November 1969 , has won a record 122 Emmy Awards in all categories ? | | |
|---|---|---|---|
| **A1:** | **Great Dane** | **A2:** | **Sesame Street** |
| P1 | The world 's most famous Great Dane first appeared on television screens on Sept. 13 , 1969 . | P1: | In its long history , Sesame Street has received more *Emmy Awards* than any other program , ... |
| P2 | premiered on broadcast television ( CBS ) Saturday morning , Sept. 13 , 1969 , ... yet beloved great Dane . | P2: | Sesame Street ... is recognized as a pioneer of the contemporary standard which combines education and entertainment in *children 's television shows* . |

Table 6: An example from Quasar-T dataset. The ground-truth answer is "*Sesame Street*". Q: question, A: answer, P: passages containing corresponding answer.

## 5 RELATED WORK

**Open Domain Question Answering**   The task of **open domain question answering** dates back to as early as (Green Jr et al., 1961) and was popularized by TREC-8 (Voorhees, 1999). The task is to produce the answer to a question by exploiting resources such as documents (Voorhees, 1999), webpages (Kwok et al., 2001) or structured knowledge bases (Berant et al., 2013; Bordes et al., 2015; Yu et al., 2017).

Recent efforts (Chen et al., 2017; Dunn et al., 2017; Dhingra et al., 2017b; Wang et al., 2017) benefit from the advances of machine reading comprehension (RC) and follow the search-and-read QA direction. These deep learning based methods usually rely on a document retrieval module to retrieve a list of passages for RC models to extract answers. As there is no passage-level annotation about which passages entail the answer, the model has to find proper ways to handle the noise introduced in the IR step. Chen et al. (2017) uses bi-gram passage index to improve the retrieval step; Dunn et al. (2017); Dhingra et al. (2017b) propose to reduce the length of the retrieved passages. Wang et al. (2017) focus more on noise reduction in the passage ranking step, in which a ranker module is jointly trained with the RC model with reinforcement learning.

To the best of our knowledge, our work is the first to improve neural open-domain QA systems by using multiple passages for evidence aggregation. Moreover, we focus on the novel problem of "text evidence aggregation", where the problem is essentially modeling the relationship between the question and multiple passages (i.e. text evidence). In contrast, previous answer re-ranking research did not address the above problem: (1) traditional QA systems like (Ferrucci et al., 2010) have similar passage retrieval process with answer candidates added to the queries. The retrieved passages were used for extracting answer scoring features, but the features were all extracted from single-passages thus did not utilize the information of union/co-occurrence of multiple passages. (2) KB-QA systems (Bast & Haussmann, 2015; Yih et al., 2015; Xu et al., 2016) sometimes use text evidence to enhance answer re-ranking, where the features are also extracted on the pair of question and a single-passage but ignored the union information among multiple passages.

**Multi-Step Approaches for Reading Comprehension**   We are the first to introduce re-ranking methods to neural open-domain QA and multi-passage RC. Meanwhile, our two-step approach shares some similarities to the previous multi-step approaches proposed for standard single-passage RC, in terms of the purposes of either using additional information or re-fining answer predictions that are not easily handled by the standard answer extraction models for RC.

On cloze-test tasks (Hermann et al., 2015), Epireader Trischler et al. (2016) relates to our work in the sense that it is a two-step extractor-reasoner model, which first extracts $K$ most probable single-token answer candidates and then constructs a hypothesis by combining each answer candidate to the question and compares the hypothesis with all the sentences in the passage. Their model differs from ours in several aspects: (i) Epireader matches a hypothesis to *every* single sentence, including all the "noisy" ones that does not contain the answer, that makes the model inappropriate for open-domain QA setting; (ii) The sentence matching is based on the sentence embedding vectors computed by a convolutional neural network, which makes it hard to distinguish redundant and complementary evidence in aggregation; (iii) Epireader passes the probabilities predicted by the extractor to the

reasoner directly to sustain differentiability, which cannot be easily adapted to our problem to handle phrases as answers or to use part of passages.

Similarly, (Cui et al., 2017) also combined answer candidates to the question to form hypotheses, and then explicitly use language models trained on documents to re-rank the hypotheses. This method benefits from the consistency between the documents and gold hypotheses (which are titles of the documents) in cloze-test datasets, but does not handle multiple evidence aggregation like our work.

S-Net (Tan et al., 2017) proposes a two-step approach for generative QA. The model first extracts an text span as the answer clue and then generates the answer according to the question, passage and the text span. Besides the different goal on answer generation instead of re-ranking like this work, their approach also differs from ours on that it extracts only one text span from a single selected passage.

# 6 CONCLUSIONS

We have observed that open-domain QA can be improved by explicitly combining evidence from multiple retrieved passages. We experimented with two types of re-rankers, one for the case where evidence is consistent and another when evidence is complementary. Both re-rankers helped to significantly improve our results individually, and even more together. Our results considerably advance the state-of-the-art on three open-domain QA datasets.

Although our proposed methods achieved some successes in modeling the union or co-occurrence of multiple passages, there are still much harder problems in open-domain QA that require reasoning and commonsense inference abilities. In future work, we will explore the above directions, and we believe that our proposed approach could be potentially generalized to these more difficult multi-passage reasoning scenarios.

# 7 ACKNOWLEDGMENTS

This work was partially supported by DSO grant DSOCL15223.

We thank Mandar Joshi for testing our model on the unfiltered TriviaQA hidden test dataset.

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
