# OpenReview forum: "Evidence Aggregation for Answer Re-Ranking in Open-Domain Question Answering"
_ICLR.cc/2018/Conference — Accept (Poster)_

### Official Review · AnonReviewer1 · 2017-11-26
**This is a neural-based approach for improving QA systems by aggregating answers from multiple passages.**

**Rating:** 6
**Confidence:** 2

**Review:**

The paper is clear, although there are many English mistakes (that should be corrected).
The proposed method aggregates answers from multiple passages in the context of QA. The new method is motivated well and departs from prior work. Experiments on three datasets show the proposed method to be notably better than several baselines (although two of the baselines, GA and BiDAF, appear tremendously weak). The analysis of the results is interesting and largely convincing, although a more dedicated error analysis or discussion of the limitation of the proposed approach would be welcome.

Minor point: in the description of Quasar-T, the IR model is described as lucene index. An index is not an IR model. Lucene is an IR system that implements various IR models. The terminology should be corrected here.

---

> ### Author Response · Authors · 2018-01-05
> **Response to ICLR 2018 Conference Paper758 AnonReviewer1**
>
> Thank you for your valuable comments! We corrected the grammar and spelling issues and revised the Lucene description on Page 6.
>
> We provided additional discussion in the conclusion section. Our analysis shows that the instances which were  incorrectly  predicted require complex reasoning and sometimes commonsense knowledge to get right.  We believe that further improvement in these areas has the potential to greatly improve performance in these difficult multi-passage reasoning scenarios.
>
> About baselines:
> The two baselines, GA and BiDAF, came from the dataset papers. Besides these two, we also compared with the R^3 baseline. This method is from the recent work (Wang et al, 2017), which improves previous state-of-the-art neural-based open-domain QA system (Chen et al., 2017) on 4 out of 5 public datasets. As a result, we believe that this baseline reflects the state-of-the-art, thus our experimental comparison is reasonable.

---

### Official Review · AnonReviewer2 · 2017-11-27
**Very good contribution on multi-sentences answer reranking with significant experimental results.**

**Rating:** 8
**Confidence:** 3

**Review:**

The authors propose an approach where they aggregate, for each candidate answer, text from supporting passages. They make use of two ranking components. A strength-based re-ranker captures how often a candidate answer would be selected while a coverage-based re-ranker aims to estimate the coverage of the question by the supporting passages. Potential answers are extracted using a machine comprehension model. A bi-LSTM model is used to estimate the coverage of the question. A weighted combination of the outputs of both components generates the final ranking (using softmax).
This article is really well written and clearly describes the proposed scheme. Their experiments clearly indicate that the combination of the two re-ranking components outperforms raw machine comprehension approaches. The paper also provides an interesting analysis of various design issues. Finally they situate the contribution with respect to some related work pertaining to open domain QA. This paper seems to me like an interesting and significant contribution.

---

> ### Author Response · Authors · 2018-01-05
> **Response to ICLR 2018 Conference Paper758 AnonReviewer2**
>
> Thank you for your kind review. We have improved the presentation and added new discussions which we hope will further improve.

---

### Official Review · AnonReviewer3 · 2017-11-29
**Incremental idea but with solid results**

**Rating:** 6
**Confidence:** 4

**Review:**

Traditional open-domain QA systems typically have two steps: passage retrieval and aggregating answers extracted from the retrieved passages.  This paper essentially follows the same paradigm, but leverages the state-of-the-art reading comprehension models for answer extraction, and develops the neural network models for the aggregating component.  Although the idea seems incremental, the experimental results do seem solid.  The paper is generally easy to follow, but in several places the presentation can be further improved.

Detailed comments/questions:
  1. In Sec. 2.2, the justification for adding H^{aq} and \bar{H}^{aq} is to downweigh the impact of stop word matching.  I feel this is a somewhat indirect and less effective design, if avoiding stop words is really the reason.  A standard preprocessing step may be better.
  2. In Sec. 2.3, it seems that the final score is just the sum of three individual normalized scores. It's not truly a "weighted" combination, where the weights are typically assumed to be tuned.
  3. Figure 3: Connecting the dots in the two subfigures on the right does not make sense.  Bar charts should be used instead.
  4. The end of Sec. 4.2: I feel it's a bad example, as the passage does not really support the answer. The fact that "Sesame Street" got picked is probably just because it's more famous.
  5. It'd be interesting to see how traditional IR answer aggregation methods perform, such as simple classifiers or heuristics by word matching (or weighted by TFIDF) and counting.  This will demonstrates the true advantages of leveraging modern NN models.

Pros:
  1. Updating a traditional open-domain QA approach with neural models
  2. Experiments demonstrate solid positive results

Cons:
  1. The idea seems incremental
  2. Presentation could be improved

---

> ### Author Response · Authors · 2018-01-05
> **Response to ICLR 2018 Conference Paper758 AnonReviewer3**
>
> Thank you for your feedback and thorough review. We have revised the paper to address the issues you raised  and fixed the presentation issues.
>
> ABOUT THE NOVELTY:
>
> Although traditional QA systems also have the answer re-ranking component, this paper focuses on a novel problem of ``text evidence aggregation'': Here the problem is essentially modeling the relationship between the question and multiple passages (i.e., text evidence), where different passages could enhance or complement each other. For example,  the proposed neural re-ranker models the complementary scenario, i.e., whether the union of different passages could cover different facts in a question, thus the attention-based model is a natural fit.
>
> In contrast, previous answer re-ranking research did not address the above problem: (1) traditional QA systems like (Ferrucci et al., 2010) used similar passage retrieval approach with answer candidates added to the queries. However they usually consider each passage individually for extracting features of answers, whereas we utilize the information of union/co-occurrence of multiple passages by composing them with neural networks. (2) KB-QA systems (Bast and Haussmann, 2015; Yih et al., 2015; Xu et al., 2016) sometimes use text evidence to help answer re-ranking, where the features are also extracted on the pair of a question and a single-passage but ignored the union information among multiple passages.
>
> We have added the above discussion to our paper (Page 11).
>
> RESPONSE TO THE DETAILED QUESTIONS:
>
> Q1: In Sec. 2.2, the justification for adding H^{aq} and \bar{H}^{aq} is to downweigh the impact of stop word matching.  I feel this is a somewhat indirect and less effective design, if avoiding stop words is really the reason.  A standard preprocessing step may be better.
>
> A1: We follow the model design in (Wang and Jiang 2017). The reason for adding H^{aq} and \bar{H}^{aq} is not only to downweigh the stop word matching, but also to take into consideration the semantic information at each position. Therefore, the sentence-level matching model (Eq. (5) in the next paragraph) could potentially learn to distinguish the effects of the element-wise comparison vectors with the original lexical information. We’ve clarified this on Page 5.
>
> Q2: In Sec. 2.3, it seems that the final score is just the sum of three individual normalized scores. It's not truly a "weighted" combination, where the weights are typically assumed to be tuned.
>
> A2: We did tune the assigned weights for the three types of normalized scores on the dev set. The tuned version gives some improvement on dev and results in slightly better test scores, compared to simply summing up the three scores.
>
> Q3: Figure 3: Connecting the dots in the two subfigures on the right does not make sense.  Bar charts should be used instead.
>
> A3: We have changed the subfigures to bar charts in the updated version.
>
> Q4: The end of Sec. 4.2: I feel it's a bad example, as the passage does not really support the answer. The fact that "Sesame Street" got picked is probably just because it's more famous.
>
> A4: We agree that the passages in Table 6 do not provide full evidence to the question (unlike the example in Figure 1b where the passages fully support all the facts in the question). However, the “Sesame Street” got picked not because it is more famous, but because it has supporting evidence in the form of the  "award-winning" and "children's TV show" facts, while the candidate "Great Dane" only covers "1969".
>
> We selected this example in order to show another common case of realistic problems in Open-Domain QA, where the question is complex and the top-K retrieved passages cannot provide full evidence. In this case, our model is able to select the candidate with evidence covering more facts in the question (i.e. the candidate that is more likely to be approximately correct).
>
>
> Q5: It'd be interesting to see how traditional IR answer aggregation methods perform, such as simple classifiers or heuristics by word matching (or weighted by TFIDF) and counting.  This will demonstrate the true advantages of leveraging modern NN models.
>
> A5: Thank you for the valuable advice! We’ve added a baseline method with BM25 value to rerank the answers based on the aggregated passages, together with the analysis about it in the current version. In summary, the BM25 model improved the F1 scores but sometimes caused a decrease in the EM scores.  This is mainly for two reasons: (1) BM25 relies on bag-of-word representation, so context information is not taken into consideration. Also it does not model the phrase similarities. (2) shorter answers are preferred by BM25. For example when answer candidate A is a subsequence of B, then according to our way of collecting pseudo passages, the pseudo passage of A is always a superset of the pseudo passage of B. Therefore F1 scores are often improved while EM declines.

---

### Decision · Program_Chairs · 2018-01-29
**ICLR 2018 Conference Acceptance Decision**

**Decision:**

Accept (Poster)

**Comment:**

The pros and cons of this paper cited by the reviewers can be summarized below:

Pros:
* Solid experimental results against strong baselines on a task of great interest
* Method presented is appropriate for the task
* Paper is presented relatively clearly, especially after revision

Cons:
* The paper is somewhat incremental. The basic idea of aggregating across multiple examples was presented in Kadlec et al. 2016, but the methodology here is different.